# Preparation and Investigation of Spherical Powder Made from Corrosion-Resistant 316L Steel with the Addition of 0.2% and 0.5% Ag

**DOI:** 10.3390/ma15227887

**Published:** 2022-11-08

**Authors:** Mikhail A. Kaplan, Artem D. Gorbenko, Alexander Y. Ivannikov, Sergey V. Konushkin, Andrey A. Kirsankin, Alexander S. Baikin, Konstantin V. Sergienko, Elena O. Nasakina, Anna V. Mikhailova, Boris A. Rumyantsev, Irina V. Gorudko, Alexey G. Kolmakov, Alexander V. Simakin, Sergey V. Gudkov, Sergey A. Oshkukov, Mikhail A. Sevostyanov

**Affiliations:** 1A.A. Baikov Institute of Metallurgy and Materials Science (IMET RAS), Russian Academy of Sciences, 119334 Moscow, Russia; 2All-Russian Research Institute of Phytopathology (VNIIF), 143050 Moscow, Russia; 3Department of Materials Science, Bauman Moscow State Technical University, 105005 Moscow, Russia; 4Department of Biophysics, Belarusian State University, 220006 Minsk, Belarus; 5Prokhorov General Physics Institute of the Russian Academy of Sciences, 119991 Moscow, Russia; 6Moscow Regional Clinical & Research Institute, 129110 Moscow, Russia

**Keywords:** corrosion-resistant steel, silver, plasma dispersion, spherical powder, granulometric composition, morphology, fluidity, bulk density, microhardness, phase composition, impurity composition

## Abstract

The paper describes the production and study of spherical powder made from corrosion-resistant 316L steel with the addition of 0.2% and 0.5% Ag. The study of granulometric composition, morphology, fluidity and bulk density, phase composition, microhardness and impurity composition of the spherical powders was carried out. The study showed compliance of the spherical powders with the requirements for powders used for additive manufacturing. The fluidity of the powders was 17.9 s, and the bulk density was 3.76 g/cm^3^. The particles have a spherical shape with a minimum number of defects and an austenitic-ferritic structure. The study of the phase composition of ingots, wires and powders showed that the ingot structure of all samples consists of austenite. According to the results of studies of the phase composition of the wire, there is a decrease in γ–Fe and an increase in α–Fe and σ–NiCr in going from wire No. 1 to wire No. 3. According to the results of studies of the phase composition of the powder particles, there are three phases, γ-Fe, α-Fe, and Fe_3_O_4_. The study of microhardness showed a decrease in HV depending on the increase in silver. The hardness of the powder is lower than that of the ingot by 16–24% due to the presence of a ferritic phase in the powder. As a result of plasma spraying, an increase in residual oxygen is observed, which is associated with the oxidation of the melt during plasma dispersion. The amount of nitrogen and sulfur does not change, while the amount of carbon and hydrogen decreases, and the impurities content corresponds to the standards for corrosion-resistant steel. Qualitative and quantitative analysis of the silver content in the samples indicates that it was not affected by the stages involved in obtaining the spherical powder.

## 1. Introduction

Corrosion-resistant steels are one of the most important materials in the world. Austenitic corrosion-resistant steels have become the most widespread and used. They have high corrosion resistance, mechanical strength and ductility. Due to this, they are used in many areas of industry [1,2,3,4]. Corrosion-resistant austenitic steel 316L has long been used for the manufacture of various surgical instruments, medical implants and jewelry [5,6,7,8,9].

However, there are limitations to the use of this material in medicine because the biological environment in the human body is aggressive for metals and this can lead to protein adsorption and corrosion and/or the appearance of a source of bacterial infection [10,11,12,13]. Silver can eliminate this negative effect when applied to the surface to form a protective coating. However, such methods are economically inefficient due to poor tribological properties. Therefore, methods of adding silver to corrosion-resistant steel are widely considered to give it antibacterial properties [14,15,16,17,18].

Austenitic corrosion-resistant steels have been actively introduced in the additive industry, in which products are created layer by layer-by-layer build-up of the material [19,20,21,22,23,24,25]. The raw material for additive manufacturing is spherical powder or wire. The main characteristics of spherical powder for 3D printing are sphericity, granulometric composition, chemical composition, fluidity and bulk density. To obtain high-quality products, spherical powders must consist of small spherical particles of a certain fraction that flow and form dense layers [26,27,28,29,30,31]. Therefore, obtaining and researching such powders from a new material is an urgent task.

There are various approaches for obtaining spherical metal powders, the most common of which are melt dispersion and spheroidization of an irregular-shaped powder. The method of spheroidization consists of the processing of powders having a non-spherical particle shape using thermal plasma. This method makes it possible to obtain spherical powders with a given granulometric composition. However, it is characterized by low performance. When the melt is dispersed, the metal or alloy is sprayed. To obtain spherical powders of stainless steels, gas atomization methods are most often used due to the high performance of the method. However, under exceptional conditions of required sphericity and roundness, powders produced using this method are not good enough to obtain high-quality products, therefore other methods are used, including plasma spraying of wire [20].

The purpose of this work is to evaluate the morphology, granulometric composition, bulk density, fluidity, phase composition and microhardness of spherical powders obtained by plasma spraying of 316L corrosion-resistant steel wire with the addition of 0.2% and 0.5% Ag.

## 2. Materials and Methods

As an object of research, three corrosion–resistant 316L steels were smelted (sample No. 1—the initial steel, sample No. 2, with 0.2% Ag added, and sample No. 3, with 0.5% Ag added). The detailed chemical composition of the three samples is presented in Table 1. Further, the spherical powder was obtained from these compositions by rolling, rotary forging, drawing to a wire with a diameter of 1 mm and further plasma spraying of the wire.

The melting of the canopies was carried out in an argon arc furnace with a non-consumable tungsten electrode LK200DI from Leybold-Heraeus (Cologne, Germany). The attachments were placed in a water-cooled copper mold, after which the working chamber was hermetically closed and vacuumed to a pressure of 1 × 10^−2^ mmHg. After that, argon was injected into the chamber to a pressure of 0.4 atm.

In the process of the first remelting, an ingot weighing 45–50 g was obtained in the form of a biconvex lens with a diameter of 30–35 mm and a height of 10–15 mm (Figure 1). The next two remelts are aimed at obtaining a uniform chemical composition over the entire volume of the ingot. The duration of each melting of one ingot was 1–1.5 min. Before melting the ingot, an ingot of titanium iodide weighing 15–20 g used as a getter was melted.

Further, under these conditions, ingots of 45–50 g are fused into single ingots weighing 180–200 g for two remelts and subjected to homogenizing annealing in a vacuum of 2 × 10^−5^ mmHg at a temperature of 1050 °C for 9 h in a vacuum furnace ESQVE-1,7.2,5/21 SHM13. The final ingot, shown in Figure 2, has a length of 90–100 mm, a width of 20–25 mm and a height of 10–15 mm.

The primary deformation of cast blanks with a height of 10–15 mm was carried out by the method of warm rolling with preheating to a temperature of 1100 °C on a two-roll mill DUO-300, with partial absolute compression per pass: 2 mm to a workpiece thickness of 10 mm. To further obtain the wire, the workpiece was rotated 90° and rolled again to the final cross-section size of 10 × 10 mm. The resulting bars are shown in Figure 3. The workpieces were heated before deformation in a KYLS 20.18.40/10 muffle furnace by HANS BEIMLER (Mexico City, Mexico), for 20–25 min before the first rolling and 5 min during intermediate heating.

To obtain a rod with a diameter of 2.4 mm from a rolled billet of 10 × 10 mm, rotary forging was carried out sequentially on radial forging machines B2129.02, B2127.01, B2123.01 with a sequential change of strikers diameters: 12.5; 11.5; 10.5; 9;5; 8.5; 7.6; 6.5; 5.8; 5; 4.7; 4.5; 3.8; 3.4; 3; 2.7; and 2.45 mm. Preheating of the workpieces in air immediately before deformation was carried out in a PTS furnace-2000-40-1200 up to 700 °C.

To change the diameter from 2.4 mm to a diameter of 1 mm, drawing was performed on a C7328/ZF machine from The Northwest Machine Co., Ltd. (Xian, Shaanxi, China). The drawing took place in cold air. The rods were etched to scale with sulfuric acid in a ratio of 1:10. Sodium soap was used as a lubricant. To ensure good adhesion of the surface of the rod and the lubricating layer, a drill (a pre-lubricating layer) was applied. Drawing from a diameter of 2.4 to a diameter of 1.6 mm was carried out with a step of 0.2 mm and a drawing speed of 5 m/min. Next, intermediate annealing was applied in a pass-through tubular electric furnace at a temperature of 900 °C for 2 min to remove the riveting and, thereby, increase plasticity. Drawing from a diameter of 1.6 to a diameter of 1 mm was carried out with a step of 0.1 mm and a drawing speed of 2.5 m/min. The resulting wire was polished and wound on a reel (Figure 4).

A spherical powder was obtained from the resulting wire with a diameter of 1 mm according to the spent mode (power of 4 kW at an electric current of 40A and a voltage of 100V and total gas consumption of 200 L/min) at a plasma spraying plant for wire created at IMET RAS (Patent No. 2749403 of the Russian Federation). In this mode, the powder is obtained with low dispersion in the granulometric composition. The spraying scheme is shown in Figure 5.

The granulometric composition was studied using an Analysette 22 NanoTec laser (Fritsch, Idar-Oberstein, Germany) diffraction particle size analyzer (Fritsch, Idar-Oberstein, Germany). Particles in the parallel laser beam scatter light at a constant solid angle, the magnitude of which depends on the diameter of the particles. The lens collects scattered light annularly on a detector installed in the focal plane of the lens. Non-scattered light always converges at the focal point on the optical axis. As a result, the diameter of the laser diffraction particle is obtained, the diameter of which is equivalent to a ball with the same distribution of scattered light. The average volume diameters are measured, and the resulting particle size distribution is a volume distribution.

The morphology of the powder particles was studied using a JEOL JSM-IT500 scanning electron microscope with a power of 15 kW at ×200 entrainment. During the study, images of the sample surface with high spatial resolution obtained in the secondary electron mode were analyzed. To study the morphology of particles in raster mode, the samples were glued onto a copper substrate using conductive carbon glue and a layer of platinum or gold was sprayed on them using the Univex300 spraying unit from Leybold (Cologne, Germany) and Fine Coat from JEOL (Tokyo, Japan).

The determination of the fluidity and bulk density of the obtained powders was carried out using a calibrated funnel (Hall device) on an HFlow-1 device in accordance with GOST 20899-98 and Part 1 of GOST 19440-94. The fluidity of the powder was determined by the time required for the expiration of 50 g of metal powder through the hole of a calibrated funnel of standardized dimensions. The bulk density was determined as the ratio of the mass of a certain amount of powder, which in a freely poured state completely fills a container of a known volume, to this volume. A freely poured state was obtained when filling the container with a funnel located above it at a certain distance. By vibrating a special vessel into which a certain amount of powdered sample was loaded, the density was determined after shaking. The vibration load was carried out at fixed values of frequency and amplitude. The result was calculated after the volume of powder in the vessel ceased to decrease. To do this, the powder mass previously determined by weighing was divided by the volume measured in the vessel.

The Vickers method was used to study microhardness. The load was 200 g/mm^2^ with a shutter speed of 10 s. The study of microhardness was carried out for samples in the form of the ingot and powder in the form of the grinds. Pressing took place on a pneumohydraulic press IP 40 at a temperature of 170 °C and an exposure time of 20 min, at a pressure of 4 atm. The obtained samples were ground, polished and measured on a microhardness meter. The hardness measurement on the particles was carried out at various diameters.

X-ray phase analysis was performed on an ARL X’TRA device (Thermo Fisher Scientific (Ecublens) SARL, Ecublens, Switzerland) with Co Kα radiation. The device was calibrated according to the standard sample NIST SRM-1976a, and the error of the position of the reflexes did not exceed 0.01° 2ϴ. The crystal lattice parameter was refined by extrapolation to ϴ = 90° by the Nelson-Riley method in the Origin-2017 program, and the microdeformation of the crystal lattice of the main phase was determined by the Williamson-Hall method in the HighScore Plus 2.0.0 (PanAnalytical) program. The quantitative content of crystal phases was estimated by the method of corundum numbers.

Determination of the content of the mass fraction of oxygen, nitrogen, carbon, sulfur and hydrogen in the samples was carried out using TC-600, RHEN-602 and CS-600 gas analyzers (Leco, St. Joseph, MI, USA). Determination of the content of the mass fraction of oxygen and nitrogen in the samples was carried out by the method of reducing melting in an inert carrier gas current, followed by detection of oxygen in the infrared cell, and nitrogen in the conductometric cell, of the Leco TC-600 gas analyzer. Helium was used as a carrier gas. When determining the hydrogen content in the samples, the method of reducing melting was also used, with high-purity argon replacing helium as the carrier gas. Hydrogen detection was carried out in the conductometric cell of the Leco RHEN-602 gas analyzer. The carbon and sulfur content was determined by the method of reducing melting in a ceramic crucible in an induction furnace and was detected by the amount of released gaseous CO_2_ and SO_2_ in the infrared cell of the CS-600 gas analyzer from Leco.

Qualitative and quantitative analysis of the elemental composition was performed on an Analytik Jena PlasmaQuant 9100 optical emission spectrometer (Jena, Germany). Optical emission spectrometers analyze solutions, so the samples were first dissolved in aqua regia (25% nitric acid and 75% hydrochloric acid) until a homogeneous solution was obtained and then tested.

## 3. Results

The analysis of the granulometric composition of powders obtained by spraying from wires with a diameter of 1 mm made from experimental compositions No. 1–3 indicates the predominance of particles less than 160 microns, which makes it possible to recommend such a powder for construction processes by additive manufacturing methods. The volume content of particles with a fractional composition corresponding to the requirements of the selective laser fusion method is less than 20%, which requires additional studies of plasma dispersion to obtain the required fractional composition (10–60 microns), however, the entire powder is suitable for additive processes by laser surfacing. The results of the granulometric composition of the obtained powders are shown in Figure 6.

The morphology of the powder particles obtained by flame atomization was studied (Figure 7a–c). The morphology of the powder particles with added silver was the same as that of the powder particles without added silver. The powder obtained by flame atomization showed high sphericity and roundness with a minimum number of defects, according to the Krubien-Schloss diagram.

The fluidity and bulk density of powders were measured by a Hall flowmeter in accordance with GOST 19440-94 and GOST 20899-98. The measurement results of the obtained powders are shown in Figure 8.

The fluidity of the powders averaged 17.9 s, and the bulk density was 3.76 g/cm^3^. These indicators meet the requirements for powders used for additive manufacturing (fluidity of 50 g of powder less than 30 s and a bulk density of more than 3 g/cm^3^).

The properties of 316L steel, including strength and ductility, can be varied in a wide range by changing its phase composition. Austenite with a face-centered cubic (FCC) lattice usually has good plasticity, while its strength is not very high [32]. This phase can be transformed into a body-centered cubic (BCC) α-phase, which has very high strength, but very limited plasticity. According to [33], plastic deformation usually leads to a martensitic phase transformation from γ-austenite to α-martensite, while the reverse phase transformation can occur during high-temperature annealing [34].

Studies of the phase composition of the sample ingots after homogenization annealing at a temperature of 1050 °C for 9 h are presented in Table 2 and Figure 9a. It is shown that the ingot structure of all samples consists of austenite, which is important for further processing and indicates the high ductility of the steel [32].

The results of the study of the phase composition of the 1-mm wire are presented in Table 3 and Figure 9b. According to the results, there is a decrease in γ–Fe and an increase in α–Fe and σ–NiCr going from the wire of composition No. 1 to the wire of composition No. 3. The presence of α–Fe ferrite and σ–NiCr intermetallide is determined by intense plastic deformation during the wire production process at temperatures not exceeding 400 °C. If the resulting wire is used in further powder production operations, additional heat treatment is not required, but if the resulting wire is used in other applications, it would be sensible to conduct additional heat treatment to obtain a single-phase structure.

The phase composition of the obtained powders was also studied (Table 4, Figure 9b). According to the results of studies of the phase composition, there are three phases, γ-Fe, α-Fe and Fe_3_O_4_. The base consists of austenite and ferrite, while a thin Fe_3_O_4_ oxide film is present on the surface, formed during the oxidation of the surface due to the presence of residual oxygen in the chamber. Therefore, X-ray phase analysis shows a high content of the oxide phase in the powders. In order to obtain a spherical powder without an oxide film, it is necessary to ensure a minimum content of residual oxygen in the spray chamber.

The impurity composition was analyzed on a 1-mm wire and the powder obtained from it. The measurement was carried out by the method of reducing and oxidative melting. The results are shown in Figure 10 and Figure 11. As a result of plasma sputtering, an increase in residual oxygen is observed, which is associated with the oxidation of the melt during plasma dispersion. The amount of nitrogen and sulfur does not change, while the amount of carbon and hydrogen decreases. The impurity composition was also analyzed on a 1-mm wire and the powder obtained from it. The measurement showed that the impurities content was reduced. The maximum permissible content for impurities in steel is: oxygen—0.2%, nitrogen—0.1%, sulfur—0.03%, and hydrogen—0.02%.

A study of the microhardness of the ingots and powder particles was carried out. The results are shown in Figure 12. The measurement results showed that, with an increase in silver in the ingot and powder particles, the hardness decreases, which is explained by the introduction of a softer component into the composition of the corrosion-resistant steel. The hardness of the powder is lower by 16–24% due to the presence of a ferritic phase in the powder.

Qualitative and quantitative analysis of the silver content in ingots, wires and powders was carried out on an optical emission spectrometer. Before the tests, the materials were dissolved in aqua regia (25% nitric acid and 75% hydrochloric acid) until a homogeneous solution was obtained. The analysis showed the presence of silver in the samples with silver added, which indicates that the process of producing the spherical powder did not affect the silver content in the samples. The results are presented in Table 5.

## 4. Conclusions

For the first time, the technology of manufacturing powders from corrosion-resistant 316L steel with the addition of 0.2% and 0.5% Ag, including ingot smelting, rolling, rotary forging, drawing, heat treatment and plasma spraying of wire, has been developed.

Analysis of the phase composition of the ingots, wire and powders showed that the ingot structure of all samples consists of austenite. In the wires, the proportion of α–Fe and σ–NiCr increases with an increase in the silver content, which is determined by the influence of thermomechanical action on the structure of the material during drawing. A phase of Fe_3_O_4_ oxide was detected in the powders after plasma spraying, which formed on the surface of the powders by the interaction of liquid droplets with residual oxygen in the spray chamber. Therefore, to increase the chemical purity of the spherical powders, it is necessary to ensure a minimum content of residual oxygen in the chamber during plasma spraying of the wire.

The analysis of the granulometric composition, morphology, fluidity and bulk density of the spherical powders showed their compliance with the requirements imposed by additive manufacturing. The fluidity of the powders was 17.9 s, and their bulk density was 3.76 g/cm^3^. The particles have a spherical shape with a minimum number of defects and an austenitic-ferritic structure. The granulometric composition is Dv 80 = 160 microns.

Microhardness analysis showed a decrease in HV with an increase in silver content. The hardness of the powder is lower than in the ingot by 16–24% due to the presence of a ferritic phase in the powder.

As a result of plasma sputtering, an increase in residual oxygen from 0.03–0.051% to 0.067–0.121% is observed, which is associated with the oxidation of the melt during plasma dispersion. The amount of nitrogen and sulfur does not change, while the amount of carbon and hydrogen decreases, which is associated with decarbonization and dehydration in the plasma jet due to high heating temperatures. The content of impurities corresponds to the standards for corrosion-resistant steel.

Qualitative and quantitative analysis of the silver content showed that in the initial ingots the silver content was 0.1979 ± 0.073% and 0.4972 ± 0.171% 0.2% and 0.5% silver, respectively, while in the spherical powders there was no significant decrease in the silver content (0.1956 ± 0.075% and 0.4851 ± 0.178%, for 0.2% and 0.5% silver, respectively). Therefore, plasma spraying in the production of spherical powders does not affect the silver content of the material.

Maintaining a given silver content is the main advantage of obtaining a powder with the proposed technology, which will provide a high level of antibacterial properties.

## Figures and Tables

**Figure 1 materials-15-07887-f001:**
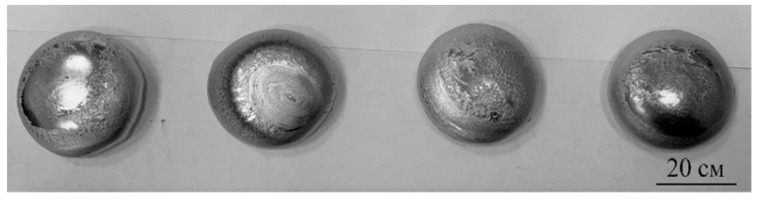
The resulting ingots weighing 45–50 g.

**Figure 2 materials-15-07887-f002:**
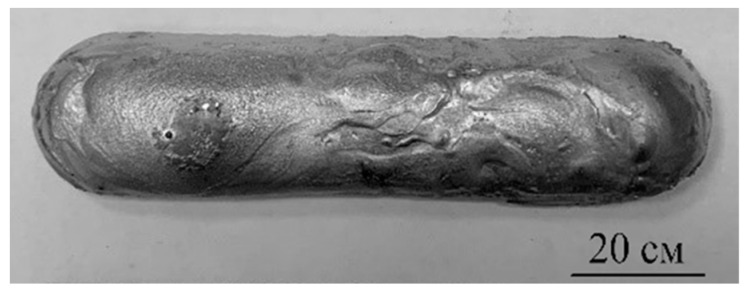
The final ingot.

**Figure 3 materials-15-07887-f003:**
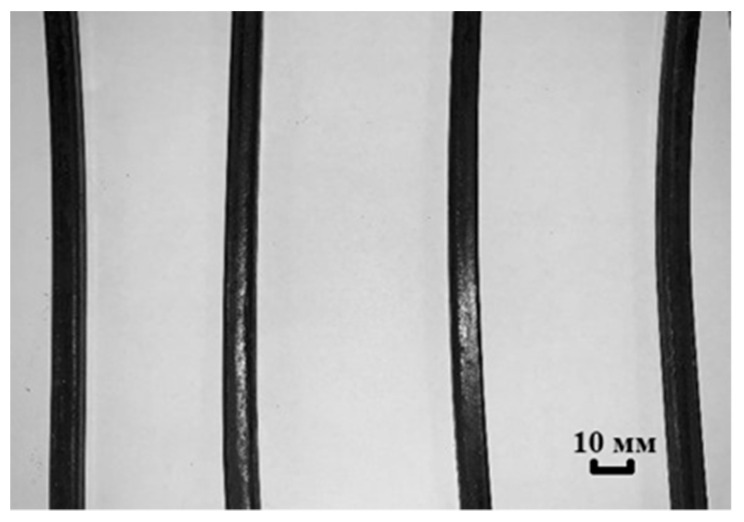
The resulting bars 10 × 10 mm.

**Figure 4 materials-15-07887-f004:**
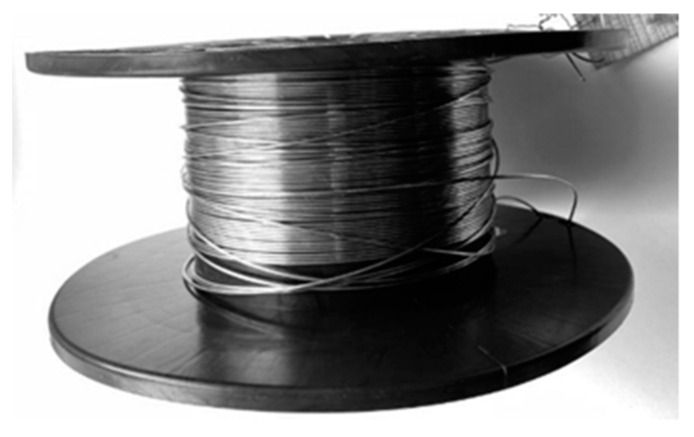
The resulting wire.

**Figure 5 materials-15-07887-f005:**
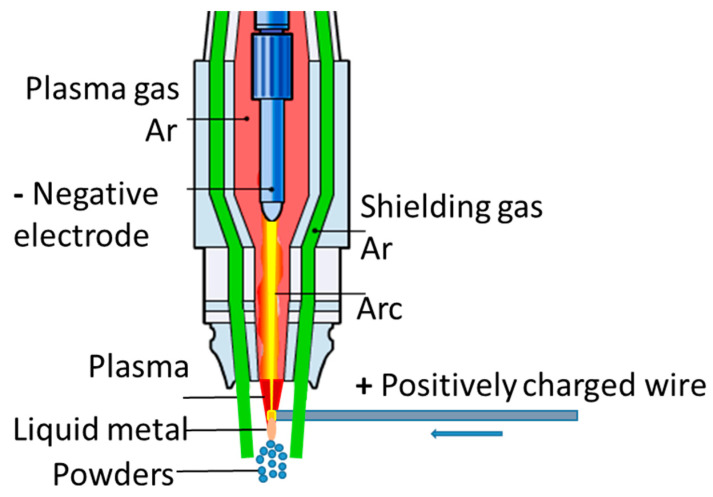
Diagram of plasma wire dispersion.

**Figure 6 materials-15-07887-f006:**
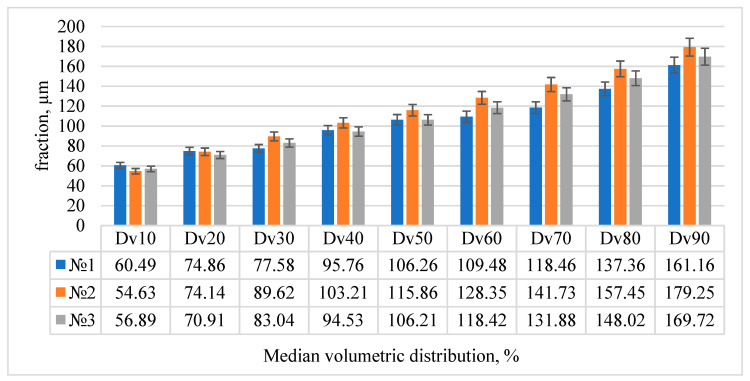
Standard percentiles of the particle size distribution in microns (Dv(n)).

**Figure 7 materials-15-07887-f007:**
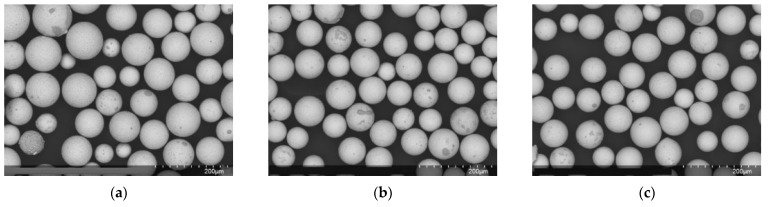
SEM of powder particles obtained by plasma atomization with a ×200 magnification (**a**)—No. 1 (316L stainless steel), (**b**)—No. 2 (+ 0.2% AG), and (**c**)—No. 3 (+0.5% silver).

**Figure 8 materials-15-07887-f008:**
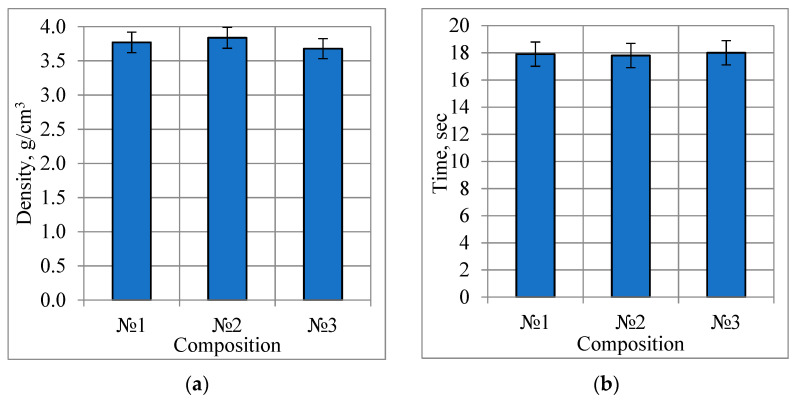
Bulk density (**a**) and fluidity (**b**) of powders.

**Figure 9 materials-15-07887-f009:**
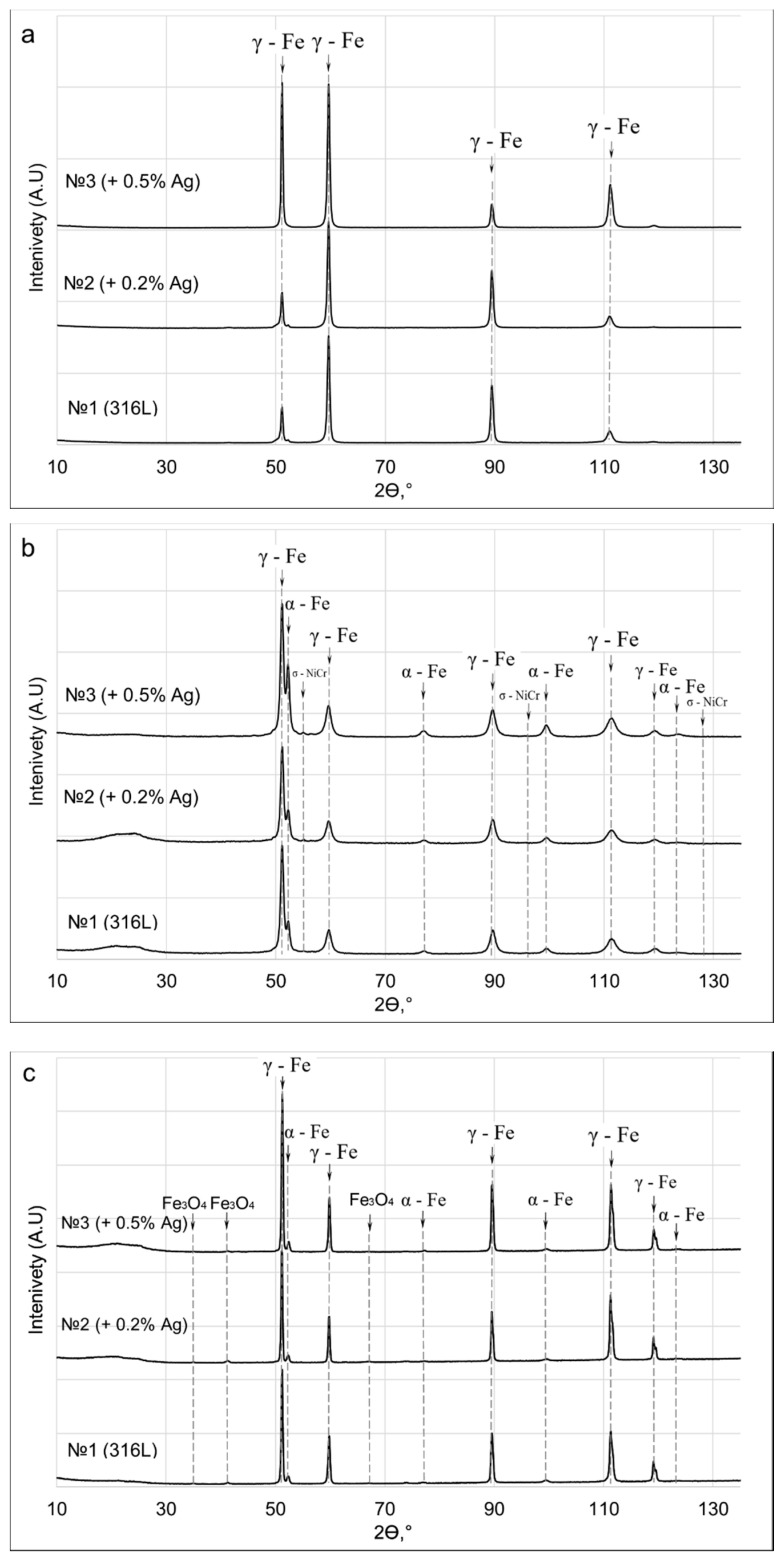
X-ray diffraction patterns of ingots (**a**), wire (**b**) and powder (**c**).

**Figure 10 materials-15-07887-f010:**
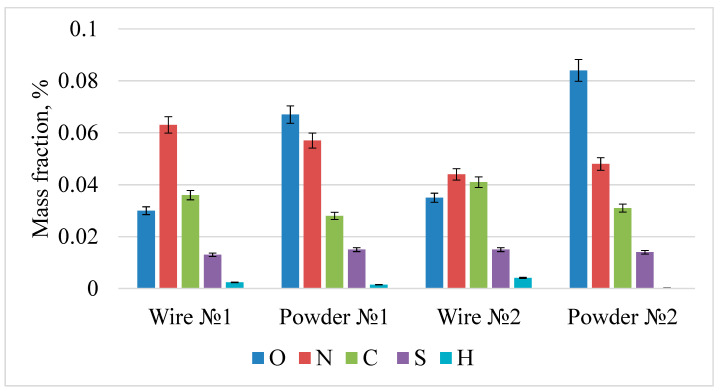
The results of the analysis of the impurity composition of samples 1 and 2.

**Figure 11 materials-15-07887-f011:**
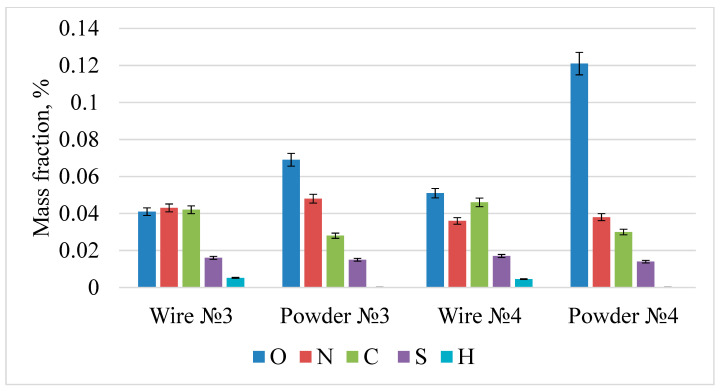
The results of the analysis of the impurity composition of samples 3 and 4.

**Figure 12 materials-15-07887-f012:**
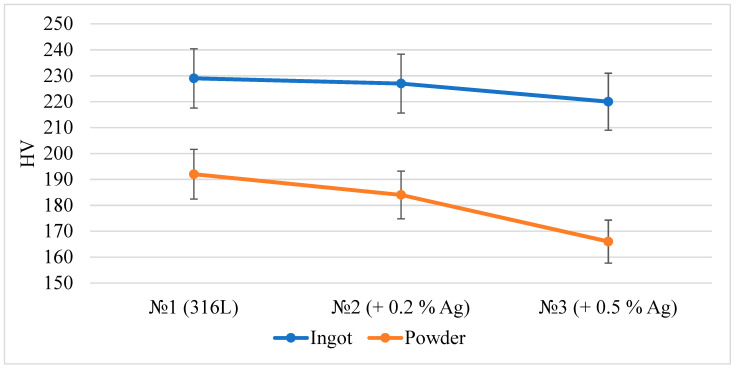
Microhardness of samples determined by the Vickers method.

**Table 1 materials-15-07887-t001:** Chemical composition of steels.

Steel	C, %	Cr, %	Ni, %	Ag, %	Si, %	Mn, %	Mo, %	Si, %
№1	0.023	17	10	0	0.5	1.5	2	0.5
№2	0.023	17	10	0.2	0.5	1.5	2	0.5
№3	0.023	17	10	0.5	0.5	1.5	2	0.5

**Table 2 materials-15-07887-t002:** Phase composition and crystal lattice parameter of ingots.

Composition	Crystal Lattice Parameters (Å)	Phase Composition	Volume Fraction, %	Weight Fraction, %
№1 (316L)	3.59605 ± 0.00004	γ–Fe	100	100
№2 (+0.2% Ag)	3.59792 ± 0.00003	γ–Fe	100	100
№3 (0.5% Ag)	3.59930 ± 0.00002	γ–Fe	100	100

**Table 3 materials-15-07887-t003:** Phase composition and parameters of the crystal lattice of 1-mm wire.

Composition	Crystal Lattice Parameters (Å)	Phase Composition	Volume Fraction, %	Weight Fraction, %
№1 (316L)	3.59442 ± 0.00008	γ–Fe	85.3 ± 0.1	85.6 ± 0.1
	2.87512 ± 0.00011	α–Fe	13.8 ± 0.1	13.5 ± 0.1
	8.81800 ± 0.00166	σ–NiCr	0.9 ± 0.1	0.8 ± 0.1
	4.57800 ± 0.00159			
№2 (+ 0.2%Ag)	3.59488 ± 0.00009	γ–Fe	82.9 ± 0.2	83.3 ± 0.2
	2.87552 ± 0.00012	α–Fe	15.8 ± 0.1	15.5 ± 0.1
	8.81800 ± 0.00168	σ–NiCr	1.3 ± 0.1	1.3 ± 0.1
	4.57800 ± 0.00163			
№3 (+ 0.5%Ag)	3.59497 ± 0.00008	γ–Fe	74.2 ± 0.1	74.7 ± 0.1
	2.87599 ± 0.00008	α–Fe	24.1 ± 0.1	23.7 ± 0.1
	8.80994 ± 0.00167	σ–NiCr	1.7 ± 0.1	1.6 ± 0.1
	4.59121 ± 0.00162			

**Table 4 materials-15-07887-t004:** Phase composition and parameters of the crystal lattice of powders.

Composition	Crystal Lattice Parameters (Å)	Phase Composition	Volume Fraction, %	Weight Fraction, %
№1 (316L)	3.59564 ± 0.00002	γ–Fe	89.7 ± 0.2	91.6 ± 0.1
	2.87621 ± 0.00012	α–Fe	4.6 ± 0.1	4.6 ± 0.1
	8.43767 ± 0.00166	Fe_3_O_4_	5.7 ± 0.2	3.7 ± 0.1
№2 (+0.2% Ag)	3.59521 ± 0.00002	γ–Fe	86.3 ± 0.2	89.4 ± 0.2
	2.87393 ± 0.00010	α–Fe	4.4 ± 0.1	4.4 ± 0.1
	8.43871 ± 0.00158	Fe_3_O_4_	9.4 ± 0.2	6.2 ± 0.1
№3 (+0.5% Ag)	3.59495 ± 0.00002	γ–Fe	92.5 ± 0.2	93.6 ± 0.1
	2.87435 ± 0.00012	α–Fe	4.7 ± 0.1	4.6 ± 0.1
	8.43300 ± 0.00165	Fe_3_O_4_	2.8 ± 0.2	1.8 ± 0.1

**Table 5 materials-15-07887-t005:** The amount of silver in the compositions (%).

Material	№1 (316L)	№2 (+0.2% Ag)	№3 (+0.5% Ag)
Ingot	0	0.1979 ± 0.073	0.4972 ± 0.171
Wire	0	0.1975 ± 0.071	0.49614 ± 0.173
Powder	0	0.1956 ± 0.075	0.4851 ± 0.178

## Data Availability

Not applicable.

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
