# Peer review of "Preparation and Investigation of Spherical Powder Made from Corrosion-Resistant 316L Steel with the Addition of 0.2% and 0.5% Ag"

_materials, 2022, doi:10.3390/ma15227887_

Round 1
Reviewer 1 Report
The results are interesting, novel and relevant, but more emphasis needs to be placed on the introduction, novelty, analysis and interpretation of the obtained results.
1. The title of this manuscript is too long. The authors should shorten it and highlight the novelty of the present study.
2. The introduction part should be extended. Various techniques are used for the production of various materials microparticles. The authors should highlight the novelty, advantages and disadvantages for various techniques. Why the doping of silver was used? How the size of produced micro-particles depend on the process parameters and etc.
3. Did the authors measure any plasma jet parameters (temperature, velocity)? Please indicate the voltage and current values of the plasma torch. The scheme of plasma torch used in the experiment should be given.
4. It is not clear what the values below the Fig.6 indicate.
5. The detailed measurements parameters of used SEM, XRD and others techniques should be included.
6. The units and notes given in the Fig. 9 is very small and should be enlarged. The authors should indicate the peak positions in the text. The values on Y axis is missing (see fig. 9, 10, 11 and 12).
7. The formation of Fe3O4 was obtained in the produced particles. It indicates a high oxidation of particles. The authors should include the elemental composition of produced particles.
8. Also, the authors need to rewrite the conclusion to make it clear how this paper significantly advances the fields and how it is different from other works.
9. No literature sources are used in the results part. The obtained results (XRD data, phase transition process, etc.) should be compared and explained using similar others authors data. This section should be extended.
10. Does the hardness depended on the size of produced powders? How the hardness was measured? Please add more detailes.
Author Response
Thank you for your useful comments and suggestions on the structure of our manuscript. Please see the attachment, there is the text of the corrected article.
- We agree with the comment. Necessary corrections were made.
New title of this manuscript is Preparation and investigation of spherical powder made from corrosion-resistant 316L steel with the addition of 0.2% and 0.5% Ag.
- We agree with the comment. Necessary corrections were made.
The introduction has been expanded. Information about getting methods was added in lines 68-78. We previously published a review article, reference 20, in detail on the methods for obtaining a spherical powder.
Silver, when added to stainless steel, gives it antibacterial properties. There is a reference to such articles 14-18. This information is also summarized in the introduction, lines 53-59.
The study of the influence of the dependence of process parameters on the size of microparticles will be carried out in a future work.
- We agree with the comment. Necessary corrections were made. A schematic of the plasma torch used in the experiment is added to Figure 5. The mode indicated on line 148 was used (power of 4 kW at an electric current of 40A and a voltage of 100V and a total gas consumption of 200 l/min)
- We agree with the comment. Necessary corrections were made. This is Standard percentiles of the particle size distribution in microns (Dv(n))
- We agree with the comment. Necessary corrections were made (expanded section of techniques, Line 157-216).
- We agree with the comment. Necessary corrections were made. Figures edited, missing values and captions added.
- We agree with the comment. Necessary corrections were made. A more detailed explanation is given in the paragraph on lines 273-280
- We agree with the comment. Necessary corrections were made. Conclusion completely rewritten and expanded (line 322-356).
- We agree with the comment. Necessary corrections were made. Added information to the methods section, lines 249-255 and 257-259
- We agree with the comment. Necessary corrections were made. The powder was obtained with a low spread in particle size distribution. Hardness was measured on particles of different diameters and the same values were obtained within the error. For this study, the methodology section on line 186-192 has also been expanded

Reviewer 2 Report
The paper presents some interesting results of preparation of spherical powders obtained by plasma spraying of 316L steel with the addition of Ag, the following questions need to be revised before publication:
1 How is the phase fraction calculated in Table 2, 3 and 4?
2 What phase does silver exist in? γ – Fe, α – Fe or σ – NiCr?
3 What is the purpose of adding silver in the article? No experimental results have been found to prove the role of silver.
4 The title of the article is too long. It is recommended to revise it.
5 The purpose of the article was not clearly stated in the Introduction section.
Author Response
Thank you for your useful comments and suggestions on the structure of our manuscript. Please see the attachment, there is the text of the corrected article.
- Necessary corrections were made. The section of methods has been expanded. Lines 157-216.
- From the state diagrams, it was revealed that 0.022% Ag is distributed between γ - Fe and α - Fe, the rest is contained in the remaining phases, in this case it is σ - NiCr (https://himikatus.ru/art/phase-diagr1/Ag- Fe.php)
- The main goal is to obtain a spherical powder from stainless steel with silver and study the composition at the stages of production. This article shows that this technology can create a spherical powder with silver in the composition without loss.
Data on the antibacterial effect were investigated in other works, which are referenced in the introduction 14-18.
- We agree with the comment. Necessary corrections were made.
New title of this manuscript is Preparation and investigation of spherical powder made from corrosion-resistant 316L steel with the addition of 0.2% and 0.5% Ag.
- We agree with the comment. Necessary corrections were made. The purpose of the article is completely rewritten, lines 79-82

Reviewer 3 Report
This paper presented the combination of mechanical and physicochemical (plasma) processes in the spheroidization of stainless steel 316L. The proposed method could be a potential technique for additive manufacturing. However, the authors should provide additional information and address the following questions:
- Line 161. Please elaborate on how the particle size and distribution were measured and analyzed. Provide any related governing equation.
- Figure 7. The morphology of the "as prepared" wire should be analyzed and compared with powder particles. Please describe how the sphericity and roundness were analyzed.
- Line 188. How were the defects measured and evaluated?
- Figure 8 (a). Were there significant differences among the three compositions? If so, why did such a trend occur?
- Line 238. Please elaborate on how to measure and analyze the impurities.
Author Response
Thank you for your useful comments and suggestions on the structure of our manuscript. Please see the attachment, there is the text of the corrected article.
- Necessary corrections were made. The section of methods has been expanded. Lines 157-216. The method for studying the granulometric composition is given on line 157-165.
- From the resulting wire with a diameter of 1 mm, a spherical powder was obtained according to the worked-out regime (power of 4 kW at an electric current of 40A and a voltage of 100V and a total gas consumption of 200 l/min) at a wire plasma spraying installation. Polished wire was used for spraying. Upon receipt, the wire is completely melted to a liquid state, so there is no direct relationship between the morphology of the wire and the morphology of the powder.
The evaluation of sphericity and roundness was carried out on the basis of electronic images of the obtained microspheres according to the Krubien-Schloss diagram.
- Defects were also assessed from the obtained images of microspheres according to the Krubien-Schloss diagram.
- No, it was not, they do not differ within the margin of error.
- Necessary corrections were made. The section of methods has been expanded. Lines 157-216. The methodology for the study of impurities is given on line 201-212.

Round 2
Reviewer 2 Report
The paper presents interesting results of preparation of spherical powders obtained by plasma spraying of 316L steel with the addition of Ag. The paper is well prepared.
Reviewer 3 Report
The revised version provided explanations for the previous comments.